# From ADHD symptoms to parental stress: The roles of functional impairment, family functioning, and parental ADHD

**Nitchawan Jongrakthanakij**, **Thanavadee Prachason**\*, **Nida Limsuwan,**
**Komsan Kiatrungrit, Masatha Thongpan, Passaporn Lorterapong,**
**Pattarabhorn Wisajun, Sudawan Jullagate**

Department of Psychiatry, Faculty of Medicine Ramathibodi Hospital, Mahidol University, Bangkok,
Thailand

\* thanavadee.pra@mahidol.edu

## Abstract

### Background

Raising a child with Attention-Deficit/Hyperactivity Disorder (ADHD) is associated with
significant parental stress. However, the complex relationships between factors in the
child and family in shaping this stress are not well understood. This study aimed to
elucidate these interrelationships and identify the key determinants of parental stress.

### Methods

A cross-sectional study included 127 children and adolescents with ADHD (70.9%
males; mean age 9.6±3.3 years) and their caregivers, recruited from the ADHD
Registry at Ramathibodi Hospital, Bangkok (2019–2023). Caregivers completed stan-
dardized measures of parental stress, child ADHD symptoms, child functional impair-
ment, family functioning, and parental ADHD symptoms. Structural equation modeling
was used to examine pathways from child and parental ADHD symptoms to parental
stress, with functional impairment and family functioning specified as mediators.

### Results

Examining child- and family-related factors separately, child ADHD symptoms
indirectly influenced parental stress via functional impairment, whereas parental
ADHD symptoms significantly influenced parental stress both directly and indi-
rectly via family functioning. In the integrated model examining both child- and
family-related factors concurrently, the direct and indirect pathways from parental
ADHD symptoms to parental stress via family functioning remained significant,
but not the pathway from child ADHD symptoms to parental stress via functional
impairment.

West Indies at Saint Augustine, TRINIDAD AND
TOBAGO

**Peer Review History:** PLOS recognizes the
benefits of transparency in the peer review
process; therefore, we enable the publication
of all of the content of peer review and
author responses alongside final, published
articles. The editorial history of this article is
available here: https://doi.org/10.1371/journal.
pone.0341817

**Data availability statement:** De-identified data supporting the findings of this study have been deposited in the Open Science Framework (OSF) and are publicly available at https://osf.io/b9r5s/. All data were anonymized to protect the identities of the individuals involved.

**Funding:** The author(s) received no specific funding for this work.

**Competing interests:** The authors have declared that no competing interests exist.

## Conclusions

Functional impairment, parental ADHD, and family functioning, rather than child ADHD symptoms, are key determinants of parental stress in families of children with ADHD. These factors should be routinely assessed and targeted to alleviate parental stress more effectively than focusing on child ADHD symptoms alone.

## Introduction

Attention-deficit/hyperactivity disorder (ADHD) is a common neurodevelopmental disorder characterized by persistent patterns of inattention, hyperactivity, and impulsivity. The global prevalence of ADHD was estimated to be 7.6% in children and 5.6% in adolescents [1]. ADHD is associated with significant functional impairments, including difficulties in peer relationships, academic underachievement, and challenges in daily activities, which not only affect the children themselves but also their families [1–3].

Parental stress is conceptualized as an imbalance between the demands of caregiving and a parent's perceived ability to cope [4]. While a certain degree of stress is inherent to the parenting role [5], chronic or excessive stress poses significant risks to the well-being of both parents and children. For parents, elevated stress is a predictor of poorer mental health outcomes, less effective parenting practices, and weakened parent-child relationships [6–10]. In children, parental stress is associated with emotional and behavioral difficulties, impaired socio-emotional development, and reduced social competence [10–13]. These findings highlight the importance of identifying modifiable contributors to parental stress that could be targeted to enhance family well-being.

Parenting a child with ADHD is associated with substantially elevated parental stress. The disorder's emotional and behavioral manifestations, ranging from frequent emotional outbursts to persistent academic difficulties, impose considerable demands on caregivers [14–19]. Previous research has consistently demonstrated that higher symptom severity is associated with greater parental stress, reduced family functioning, and increased caregiver burden [15,20–24]. Additionally, related functional impairments, such as noncompliance, organizational difficulties, and sustained attention deficits, further exacerbate caregiver stress [25].

Beyond child-related factors, family functioning has been reported to be another critical determinant of parental stress. Families of children with ADHD frequently exhibit reduced cohesion, strained parent-child interactions, and elevated marital conflict [26–28]. Due to a high heritability of ADHD, such family dynamics are usually compounded by the presence of parental ADHD symptoms that frequently contribute to occupational instability and interpersonal problems [8,26,29–31], creating a more complex and stressful family environment that could be passed on from generation to generation.

Although prior research [8,32–34] has examined the influence of child and parental ADHD symptoms, functional impairment, and family functioning on parental stress, these factors have often been studied in isolation. Their combined effects, particularly

through potential mediating pathways, remain poorly understood. The present study aimed to address this gap by applying structural equation modeling (SEM) to examine the contributions of both child- and family-related factors to parental stress within families affected by ADHD. For child-related factors, we hypothesized that child functional impairment would be a mediator between child ADHD symptoms and parental stress. For family-related factors, we hypothesized that family functioning would mediate the pathway from parental ADHD symptoms to parental stress. To examine the joint effects, both child- and family-related factors were integrated in the final path model to parental stress. Such insights would be a stepping stone to developing targeted, evidence-based interventions to mitigate parental stress and optimize treatment outcomes, not only for children with ADHD, but also for the entire family unit.

## Materials and methods

### Study design and participants

This cross-sectional study utilized baseline data from the ADHD Registry of the Department of Psychiatry at Ramathibodi Hospital, Bangkok, Thailand, a longitudinal cohort that consecutively enrolled children and adolescents suspected of having ADHD who first visited the psychiatric outpatient unit from October 2019 to October 2023. Children and adolescents who were unable to communicate in Thai were excluded. After informed written assent and consent were obtained from the participants and their parents, respectively, child and adolescent psychiatrists or psychiatry residents performed comprehensive clinical interviews with both the children and their caregivers to determine the diagnoses for the presenting problems as part of routine clinical care. A checklist of the DSM-5 criteria for ADHD [35] was recorded. The diagnosis of ADHD was defined as the presence of ≥6 symptoms in either the inattention or hyperactivity/impulsivity domains, beginning before age 12, present in at least two settings, and causing functional impairment. Children were excluded if symptoms occurred only during a psychotic disorder or were better explained by another psychiatric condition or substance use. Comorbidity identified at the initial visit was also recorded. Caregivers provided demographic and personal information (age, sex, living arrangements, marital status, education level, and family monthly income) and completed standardized questionnaires assessing parental stress, child ADHD symptoms, child functional impairment, family functioning, and parental ADHD symptoms. Multiple follow-up assessments were conducted over a one-year period after the initial visit. The Ethics Committee of Ramathibodi Hospital, Mahidol University, approved the data collection protocol of the registry (MURA2019/815).

For the present study, we retrieved anonymized baseline data from the ADHD registry comprising 164 children and adolescents who met the DSM-5 diagnostic criteria of ADHD. Only participants with available data on parental stress were included in this study because it was the primary outcome variable in the structural equation modeling. The sample with over 20% missing items of any questionnaire used in the models was excluded from the related analysis. The final sample for the main analyses comprises 127 participants. The characteristics of the excluded sample were not significantly different from the included sample (S1 Table). The Ethics Committee of Ramathibodi Hospital, Mahidol University, waived the requirement for informed consent to use the de-identified data for this study (MURA2025/340).

### Measures

**Parental stress.** The Perceived Stress Scale–10 (PSS-10) is a 10-item self-report instrument assessing the degree to which individuals perceive their lives as stressful [36]. Respondents rate how often they have experienced specific thoughts or feelings over the past month on a 5-point Likert scale (0 = never to 4 = very often). Total scores range from 0 to 40, with higher scores indicating greater perceived stress. The Thai version PSS-10 has demonstrated good reliability and validity in Thai populations [37] and demonstrated acceptable internal consistency in the present study, with a Cronbach's alpha of 0.73.

**Child ADHD symptoms.** The Swanson, Nolan, and Pelham IV Scale (SNAP-IV) is a 26-item parent-report and teacher-report measure used to assess ADHD symptoms of inattention and hyperactivity/impulsivity, as well

as oppositional defiant disorder (ODD) symptoms [38,39]. Each item is rated on a four-point scale, from 0 (not at all) to 3 (very much). A composite ADHD symptom score was derived by summing the inattentive and hyperactive/impulsive subscales, with higher scores indicating greater severity. The SNAP-IV is widely used in clinical and research settings to screen baseline symptoms, monitor treatment progress, and evaluate outcomes [40]. The Thai version has demonstrated strong validity and reliability [41]. In this study, internal consistency was excellent (Cronbach's alpha = 0.92).

**Child's functional impairments.** The Weiss Functional Impairment Rating Scale-Parent (WFIRS-P) is a 50-item parent-rated instrument designed to assess the impact of ADHD symptoms on a child's daily functioning [42]. Items are rated on a 4-point Likert scale (0 = not at all/never to 3 = very much/very often), with an additional "not applicable" option. Higher total scores indicate greater functional impairment. The Thai version of the WFIRS-P has demonstrated strong reliability and validity, [43]. In this study, internal consistency was excellent (Cronbach's alpha = 0.91).

**Family functioning.** The Systemic Clinical Outcome and Routine Evaluation (SCORE-15) is a 15-item self-report questionnaire for family members aged 12 years and older. Items are rated on a 5-point Likert scale (1 = "describes us very well" to 5 = "describes us not at all"), with higher scores indicating poorer family functioning [44]. The SCORE-15 has demonstrated good reliability and validity in the Thai population [45,46]. In this study, internal consistency was good (Cronbach's alpha = 0.88).

**Parental ADHD.** The Adult ADHD Self-Report Scale Screener V1.1 (ASRS-v1.1) is an 18-item self-report instrument assessing ADHD symptoms in adults [47–49]. Items are rated on a 5-point Likert scale from "never" to "very often," with total scores ranging from 0 to 72; higher scores indicate greater symptom severity. The Thai version ASRS-v1.1 has been shown to be psychometrically reliable and valid for screening adult ADHD in Thai populations [50]. In this study, internal consistency was excellent (Cronbach's alpha = 0.94).

## Statistical analysis

All analyses were conducted using Stata version 18. To maximize the power of analysis, we calculated the PSS-10, SNAP-IV, WFIRS-P, SCORE-15, and ASRS-v1.1 scores by averaging all available items in each questionnaire if the number of missing items did not exceed 20%. This approach ensures that only valid data are used for scoring while minimizing potential bias due to missing information. Sample characteristics were summarized using descriptive statistics. Spearman correlation analyses were performed to examine relationships between parental stress (PSS-10), child ADHD symptoms (SNAP-IV), functional impairments (WFIRS-P), family functioning (SCORE-15), and parental ADHD symptoms (ASRS-v1.1). To examine pathways from parental and child ADHD symptoms to parental stress, structural equation modeling (SEM) was employed. Three mediated linear regression models were tested, with parental stress as the dependent variable. First, to examine the influence of child-related factors, child ADHD symptoms were specified as the independent variable, and child functional impairment as the mediator. Second, to examine the influence of family-related factors, parental ADHD symptom was specified as the independent variable, and family functioning as the mediator. Finally, to examine the joint influences of child- and family-related factors, the first two models were combined, with child and parental ADHD symptoms as the independent variables, and child functional impairment and family functioning as the mediators. As family functioning might also influence perceived functional impairment [51], we added a mediating path from family functioning to child functional impairment to the model. Standardized z-scores were used to facilitate direct comparisons between variables. Model fit was evaluated using the Comparative Fit Index (CFI), the Root Mean Square Error of Approximation (RMSEA), and the Standardized Root Mean Square Residual (SRMR). Acceptable fit was defined as CFI ≥ 0.95, RMSEA ≤ 0.08, and SRMR ≤ 0.08 [52–54]. Primary analyses were adjusted for the child's age and sex. To assess the robustness of the findings, multiple sensitivity analyses were performed: 1) additionally controlling for family income or comorbid specific learning disorder, or 2) using only participants with complete questionnaire data. Statistical significance was set at $p < 0.05$ (two-tailed).

 

## Results

### Sample characteristics

The sample comprised 90 males (70.9%) and 37 females (20.1%), with mean age of 9.6 ± 3.3 years. ADHD presentations were predominantly inattentive (n = 56, 44.1%) or combined (n = 55, 43.3%), with hyperactive-impulsive presentation being the least common (n = 16, 12.6%). Caregivers were primarily mothers (n = 96, 81.4%), and most children lived with married parents (n = 66, 58.9%). The majority of parents held a bachelor's degree or higher (mothers: n = 88, 74.0%; fathers: n = 68, 60.2%), and the median family monthly income ranged from 50,001–100,000 baht (approximately $1,550–$3,100 USD). Detailed demographic characteristics are presented in Table 1.

### Correlations between parental stress and child- and family-related factors

Parental stress was most strongly correlated with family dysfunction (r = 0.51, p < 0.001), followed by parental ADHD symptoms (r = 0.41, p < 0.001) and child functional impairments (r = 0.39, p < 0.001). Child ADHD symptoms did not significantly correlate with parental stress (r = 0.16, p = 0.065). The correlations among the other variables were also noteworthy (Table 2). Child ADHD symptoms were strongly correlated with functional impairments (r = 0.53, p < 0.001). In turn, functional impairments showed a moderate correlation with family functioning (r = 0.46, p < 0.001). Poor family functioning was also significantly positively correlated with both parental ADHD symptoms (r = 0.39, p < 0.001) and child ADHD symptoms (r = 0.20, p = 0.023).

### Pathway from child ADHD symptoms to parental stress

As shown in Fig 1, child ADHD symptoms was not directly associated with parental stress (β [95% CI] = −0.05 [−0.26, 0.16], p = 0.634) but indirectly had an influence on parental stress via functional impairment (indirect effect β [95% CI] = 0.24 [0.11, 0.38], p < 0.001). Sensitivity analyses additionally controlling for family income or comorbid specific learning disorder confirmed the significant indirect effect of child ADHD symptoms on parental stress via functional impairment (S1 and S2 Figs). Analyses restricted to participants with complete data yielded consistent findings (S3 Fig).

### Pathway from parental ADHD symptoms to parental stress

Parental ADHD symptoms significantly exerted an influence on parental stress both directly (β [95% CI] = 0.20 [0.05, 0.35], p = 0.010) and indirectly via family functioning (indirect effect β [95% CI] = 0.15 [0.06, 0.24], p = 0.001) (Fig 2). Sensitivity analyses, additionally adjusting for family income and comorbid specific learning disorder, confirmed that both the direct and indirect effects remained significant (S4 and S5 Figs). Analyses restricted to participants with complete data produced consistent results (S6 Fig).

### Integrated pathway from ADHD symptoms to parental stress

In the final model incorporating both child- and family-related factors (Fig 3), parental ADHD symptoms exerted a significant influence on parental stress both directly (direct effect β [95% CI] = 0.18 [0.02, 0.33], p = 0.027) and indirectly via family functioning (indirect effect β [95% CI] = 0.12 [0.03, 0.20], p = 0.006), but not via child functional impairment (indirect effect β [95% CI] = 0.02 [−0.01, 0.05], p = 0.273). Child ADHD symptoms did not significantly influence parental stress neither directly (β [95% CI] = −0.04 [−0.22, 0.15], p = 0.690) nor indirectly via child functional impairment (indirect effect β [95% CI] = 0.06 [−0.03, 0.16], p = 0.208) or family functioning (indirect effect β [95% CI] = 0.07 [−0.01, 0.15], p = 0.090). Family functioning also had a significant influence on child functional impairment (β [95% CI] = 0.33 [0.19, 0.46], p < 0.001) but did not contribute to parental stress through this pathway (indirect effect β [95% CI] = 0.04 [−0.02, 0.11], p = 0.216) (Fig 3).

In sensitivity analyses, all models confirmed a significant indirect effect of parental ADHD symptoms on parental stress via family functioning, but not child functional impairment (S7–S9 Figs). However, the significant direct effect of parental

**Table 1. Sample characteristics.**

| Variable | Total N | n (%) |
|---|---|---|
| Child's age, Mean (SD) | 127 | 9.59 (3.3) |
| Male | 127 | 90 (70.9) |
| Informant | 118 | |
| Mother | | 96 (81.4) |
| Father | | 15 (12.7) |
| Relatives | | 7 (5.9) |
| Living with | 112 | |
| Father and mother | | 66 (58.9) |
| Mother | | 34 (30.4) |
| Father | | 4 (3.6) |
| Other | | 8 (7.1) |
| Parental marital status | 111 | |
| Married | | 70 (63.1) |
| Divorced | | 22 (19.8) |
| Separated | | 18 (16.2) |
| Widowed | | 1 (0.9) |
| Paternal age, Mean (SD) | 106 | 42.09 (8.4) |
| Paternal educational level | 113 | |
| Pre-university education | | 16 (14.2) |
| Vocational Certificate/High Vocational Certificate/Diploma | | 24 (21.2) |
| Bachelor's degree | | 54 (47.8) |
| Postgraduate degree | | 14 (12.4) |
| Unknown | | 5 (4.4) |
| Maternal age, Mean (SD) | 112 | 39.36 (6.5) |
| Maternal educational Level | 119 | |
| Pre-university education | | 18 (15.1) |
| Vocational Certificate/High Vocational Certificate/Diploma | | 13 (10.9) |
| Bachelor's degree | | 69 (58.0) |
| Postgraduate degree | | 19 (16.0) |
| Family monthly income | 111 | |
| <15,000 baht | | 7 (6.3) |
| 15,001–25,000 baht | | 12 (10.8) |
| 25,001–50,000 baht | | 30 (27.0) |
| 50,001–100,000 baht | | 46 (41.4) |
| >100,000 baht | | 16 (14.4) |
| ADHD DSM-5 criteria diagnosis | 127 | |
| Inattention presentation | | 56 (44.1) |
| H/I presentation | | 16 (12.6) |
| Combined type | | 55 (43.3) |
| Comorbidity | | |
| SLD | 117 | 26 (22.2) |
| ODD | 117 | 5 (4.3) |
| ASD | 117 | 3 (2.6) |
| Anxiety disorder | 116 | 3 (2.6) |
| Tic disorders | 116 | 2 (1.7) |
| Mood disorder | 117 | 1 (0.9) |

*(Continued)*

**Table 1.** (Continued)

| Variable | Total N | n (%) |
|---|---|---|
| SUD | 118 | 1 (0.9) |

**Abbreviations:** ADHD, Attention deficit hyperactivity disorder; H/I, Hyperactivity and impulsivity; SLD, Specific learning disorders; ODD, Oppositional defiant disorder; ASD, Autism spectrum disorder; SUD, Substance use disorder.

**Table 2. Spearman correlations among parental stress, child ADHD symptoms, child's functional impairments, family function, and parental ADHD symptoms (N = 127).**

| | Mean (SD) | Parental stress | Child ADHD symptoms | Child's functional impairments | Family function | Parental ADHD |
|---|---|---|---|---|---|---|
| **Parental stress** | 1.79 (0.53) | 1.000 | | | | |
| **Child ADHD symptoms** | 1.76 (0.50) | 0.16 | 1.000 | | | |
| **Child's functional impairments** | 0.79 (0.36) | 0.39*** | 0.53*** | 1.000 | | |
| **Family function** | 2.35 (0.60) | 0.51*** | 0.25** | 0.46*** | 1.000 | |
| **Parental ADHD** | 1.13 (0.77) | 0.41*** | 0.20* | 0.37*** | 0.39*** | 1.000 |

Abbreviations: ADHD, Attention deficit hyperactivity disorder.

Note: *p < 0.05; **p < 0.01; ***p < 0.001.

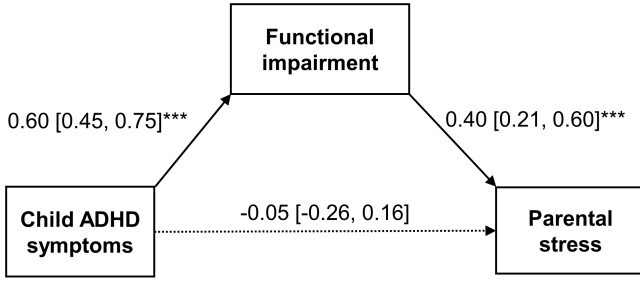

**Fig 1. Path analysis model assessing the effect of child ADHD symptoms on parental stress.** The model was mediated by functional impairment and adjusted for child age and sex. Model fit indices: $\chi^2(7) = 71.360$, $p < 0.001$; Comparative Fit Index (CFI) = 1.000; Root Mean Square Error of Approximation (RMSEA) < 0.001; Standardized Root Mean Square Residual (SRMR) < 0.001. Significant paths are indicated by *p < 0.05, **p < 0.01, and ***p < 0.001. Sample size (N) = 127.

ADHD was only confirmed in the model additionally adjusted for specific learning disorder (S8 Fig). The insignificant direct and indirect effects of child ADHD symptoms on parental stress through functional impairment remained consistent across all sensitivity analyses (S7–S9 Figs), but the indirect effect of child ADHD symptoms on parental stress through family functioning became significant in the model additionally adjusted for specific learning disorder (S8 Fig).

## Discussion

This study aimed to elucidate the influences of child-related and family-related factors on parental stress in a clinical sample of children with ADHD. We hypothesized that child functional impairment would be a mediator between child ADHD symptoms and parental stress, whereas family functioning would be a mediator between parental ADHD symptoms and parental stress. The findings partially supported our hypotheses. Specifically, child ADHD symptoms indirectly influenced

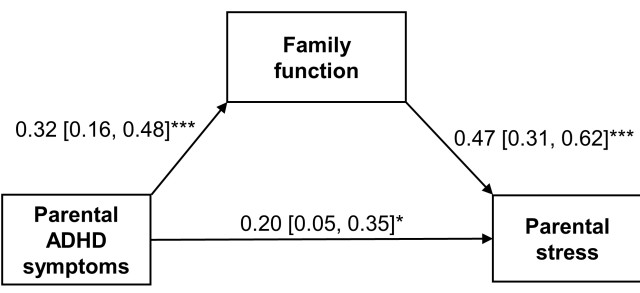

**Fig 2. Path analysis model assessing the effect of parental ADHD symptoms on parental stress.** The model was mediated by family function and adjusted for child's age and sex. The model fit indices are as follows: $\chi^2(7) = 65.723$, $p < 0.001$; Comparative Fit Index (CFI) = 1.000; Root Mean Square Error of Approximation (RMSEA) < 0.001; Standardized Root Mean Square Residual (SRMR) < 0.001. Significant paths are indicated by *$p < 0.05$, **$p < 0.01$, and ***$p < 0.001$. Sample size (N) = 127.

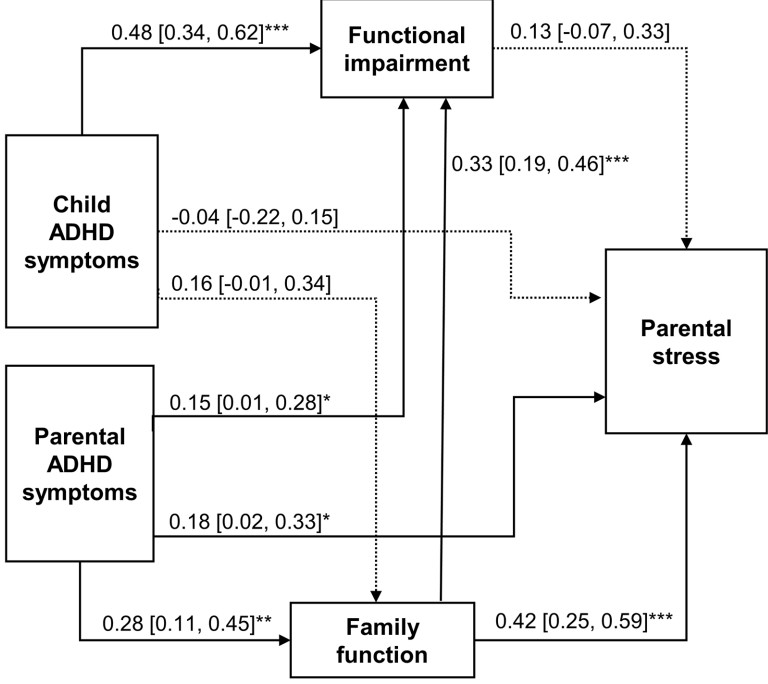

**Indirect pathway**

| | |
|---|---|
| - Child ADHD symptoms → Functional impairment → Parental stress | 0.06 [-0.03, 0.16] |
| - Child ADHD symptoms → Family function → Parental stress | 0.07 [-0.10, 0.15] |
| - Parental ADHD symptoms → Functional impairment → Parental stress | 0.02 [-0.01, 0.05] |
| - Parental ADHD symptoms → Family function → Parental stress | 0.12 [0.03, 0.20]** |

**Fig 3. Path analysis model assessing the effects of child and parental ADHD symptoms on parental stress.** The model was mediated by child functional impairment and family functioning and adjusted for child age and sex. Model fit indices were as follows: $\chi^2(15) = 153.471$, $p < 0.001$; Comparative Fit Index (CFI) = 1.000; Root Mean Square Error of Approximation (RMSEA) < 0.001; and Standardized Root Mean Square Residual (SRMR) < 0.001. Significant paths are indicated as *$p < 0.05$, **$p < 0.01$, and ***$p < 0.001$. Sample size (N) = 127.

parental stress through child functional impairment, whereas parental ADHD symptoms both directly and indirectly influenced parental stress via family functioning. The mediated pathways remained significant across sensitivity analyses, indicating robustness of the findings. However, when considered concurrently, child functional impairment was no longer significant in mediating the relationship between child ADHD symptoms and parental stress, but parental ADHD symptoms remained directly and indirectly influencing parental stress through family functioning, underscoring the role of family-related factors in shaping perceived stress in parents of children with ADHD.

Focusing on child-related factors, we found that child ADHD symptoms were associated with parental stress primarily through functional impairment rather than a direct pathway. This finding aligns with prior research demonstrating that the impact of a child's ADHD symptoms on parental well-being is often mediated by functional difficulties, including emotional dysregulation, executive functioning deficits, and social or academic challenges [32,33]. Compared to neurotypical peers, children with ADHD face more pronounced impairments in academic performance, peer interactions, and emotional regulation, which consequently increase caregiver burden. Furthermore, parents may experience social discomfort and heightened vigilance in public settings due to their child's unpredictable behaviors [33,55,56]. Such problems may not be fully resolved solely through medical treatment and require psychological and behavioral interventions to improve the child's self-regulation, social skills, and adaptive functioning within the family environment [57–59]. Recent guidelines and reviews emphasize that optimal ADHD management should target both symptoms and functional outcomes [60,61], yet functional impairment is still commonly under-assessed and under-treated in practice. Together with our findings, these pieces of evidence suggest that functional difficulties across multiple settings in the child's life should be assessed and targeted in routine clinical care of children with ADHD to improve the well-being of both children and their parents.

Focusing on family-related factors, we found that both parental ADHD symptoms and poorer family functioning were significantly associated with elevated parental stress, consistent with previous reports [22,34,62,63]. Additionally, our path analysis revealed that, apart from its direct effect, parental ADHD symptoms also exerted an indirect influence on parental stress. Given the high heritability of ADHD, parents of children with the disorder commonly exhibit elevated ADHD symptoms themselves [64–66]. These difficulties—spanning attention, planning, organization, and emotion regulation—can compromise a parent's ability to maintain routines, implement consistent parenting, and adhere to treatment protocols [42,63]. Even subclinical parental ADHD symptoms have been linked to household disorganization and heightened stress [67]. When both parent and child have ADHD, the compounding challenges lead to disrupted routines, reduced intervention efficacy, and increased emotional strain [8,29,68]. The literature also suggests that maternal ADHD, in particular, may be associated with lower use of positive parenting strategies and greater reliance on reactive behaviors, which can weaken secure attachment and create a chaotic home environment that exacerbates child behavioral problems [68–71]. Therefore, screening and managing parental ADHD may be a significant game-changer in reducing caregiver stress and facilitating the implementation of positive parenting strategies, which are essential for supporting optimal developmental outcomes in children with ADHD.

To reflect the interactive nature of a family system, we examined the joint influences of both child- and family-related factors on parental stress in an integrated path model. Consistent with the independent analysis of family-related factors, family functioning not only directly influenced parental stress but also mediated the relationship between parental ADHD symptoms and parental stress. On the contrary, child functional impairment lost its significant direct influence on parental stress and, consequently, no longer mediated the relationship between child ADHD symptoms and parental stress. These findings imply that, in a family system, parental ADHD and the broader quality of family functioning are more influential than child-related factors in shaping parental stress in families affected with ADHD. Prior research has shown that families of children with ADHD often exhibit lower cohesion, poorer communication, higher conflict, and reduced household organization, all of which are known to amplify caregiver stress [34,72–74]. Consequently, interparental conflict, especially concerning child-rearing and behavioral management, can contribute to inconsistent parenting, further aggravating child behavioral problems and marital discord [65]. These difficulties are intensified when a parent also has ADHD [75–79]. Conversely,

engaging in positive family processes, including adaptability, effective communication, and reflective capacity, may serve to mitigate parental stress and promote resilience in caregivers [73]. Taken together, these findings emphasize the importance of family functioning quality and systemic perspectives in managing parental stress in families impacted by ADHD.

### Clinical implications

The present findings provide empirical support that child-focused interventions alone may be insufficient to substantially alleviate parental stress in families of children with ADHD. Instead, a more comprehensive approach that addresses both child- and family-related factors, including parental ADHD symptoms and family functioning, appears critical for optimizing family outcomes. Prior evidence suggests that interventions aimed at improving the quality and efficiency of parent–child interactions, thereby enhancing mutual connectedness, can reduce parental stress [80]. In this regard, parent-focused interventions warrant particular attention. For example, the Improving Parenting Skills for Adults with ADHD (IPSA) program—a manualized, group-based intervention specifically tailored for parents with ADHD—has demonstrated preliminary acceptability, accessibility, and safety, with promising benefits such as enhanced parental self-efficacy, reduced stress, and decreased household chaos [81]. Therefore, integrative therapeutic interventions aiming at managing difficulties in both parents and children in a family system are critically needed to improve the well-being of families affected by ADHD.

### Strengths and limitations

A major strength of this study lies in its integrative framework, which simultaneously examined child- and family-related factors to clarify their contributions to parental stress. Using pathway analysis, we delineated the direct and indirect influences of child and parental ADHD symptoms, providing a deeper understanding of the mechanisms underlying parental stress. These findings offer valuable insights into the multifactorial contributors to parental stress in families affected by ADHD and underscore the importance of interventions that address both child and parental ADHD symptoms and family-level factors.

Several limitations of the present study should be acknowledged. First, our cross-sectional design precludes causal inference, highlighting the need for longitudinal research to clarify the direction and temporal dynamics of the observed associations. Second, as this study aimed to explore the interrelationships between factors associated with parental stress within families affected by ADHD, we did not include neurotypical children as a control group, which limits the comparison of our findings with normative levels of family functioning and parental stress. Third, the registry data used in this study were derived from a single assessment of routine psychiatric service, normally lasting 60–90 minutes, with assurance of comprehensive clinical evaluation for ADHD diagnosis. Therefore, comorbidities that were not evident at the initial encounter may be overlooked or inadequately evaluated due to the time limit. Indeed, the prevalence of comorbid conditions commonly co-occurring with ADHD was low in our dataset, except for specific learning disorder. For this reason, we only additionally adjusted for specific learning disorder in our sensitivity analyses to confirm the robustness of the findings, rather than including it as a separate predictor in the models. Fourth, participants were recruited from a university hospital, and the sample predominantly comprised male children and their mothers from highly educated families, which may limit the generalizability of the findings. Therefore, future studies should recruit larger, diagnostically diverse samples and examine the independent and combined effects of ADHD, as well as common comorbid conditions, on family functioning and parental stress. Finally, the reliance on parent-reported questionnaires introduces the potential for reporter bias. Incorporating multiple informants, including fathers and children's self-reports, would enable a more comprehensive assessment of family functioning and parental stress.

### Conclusion

This study underscores the dominant role of family-related factors, particularly parental ADHD symptoms and family functioning, in shaping stress among parents of children with ADHD. While child ADHD symptoms indirectly influence parental

stress through functional impairments, parental ADHD symptoms influence parental stress both directly and indirectly via family functioning, which remained significant even when considering both child- and family-related factors simultaneously. These findings provide empirical support for a systemic approach to target not only child symptoms but also associated functional impairment, parental ADHD, and family functioning to improve the well-being of families impacted by ADHD.

## Supporting information

**S1 Table. Characteristics of the included and excluded samples in this study.**
(DOCX)

**S1 Fig. Sensitivity analysis additionally adjusted for family income, assessing the paths from child ADHD symptoms to parental stress.**
(DOCX)

**S2 Fig. Sensitivity analysis additionally adjusted for comorbid specific learning disorder, assessing the paths from child ADHD symptoms to parental stress.**
(DOCX)

**S3 Fig. Sensitivity analysis in a sample with complete data, assessing the paths from child ADHD symptoms to parental stress.**
(DOCX)

**S4 Fig. Sensitivity analysis additionally adjusted for family income, assessing the paths from parental ADHD symptoms to parental stress.**
(DOCX)

**S5 Fig. Sensitivity analysis additionally adjusted for comorbid specific learning disorder, assessing the paths from parental ADHD symptoms to parental stress.**
(DOCX)

**S6 Fig. Sensitivity analysis using a sample with complete data, assessing the paths from parental ADHD symptoms to parental stress.**
(DOCX)

**S7 Fig. Sensitivity analysis additionally adjusted for family income, assessing the paths from child and parental ADHD symptoms to parental stress.**
(DOCX)

**S8 Fig. Sensitivity analysis additionally adjusted for comorbid specific learning disorder, assessing the paths from child and parental ADHD symptoms to parental stress.**
(DOCX)

**S9 Fig. Sensitivity analysis using a sample with complete data, assessing the paths from child and parental ADHD symptoms to parental stress.**
(DOCX)

## Acknowledgments

We express our gratitude to Assoc. Prof. Nida Limsuwan for granting access to the ADHD Registry and to the children and families at Ramathibodi Hospital, Bangkok, for their participation, which made this study possible.

## Author contributions

**Conceptualization:** Thanavadee Prachason, Komsan Kiatrungrit, Masatha Thongpan, Passaporn Lorterapong, Nida Limsuwan.

**Data curation:** Thanavadee Prachason, Komsan Kiatrungrit, Masatha Thongpan, Passaporn Lorterapong, Pattarabhorn Wisajun, Sudawan Jullagate, Nida Limsuwan.

**Formal analysis:** Nitchawan Jongrakthanakij, Thanavadee Prachason.

**Investigation:** Nitchawan Jongrakthanakij, Thanavadee Prachason, Masatha Thongpan, Nida Limsuwan.

**Methodology:** Nitchawan Jongrakthanakij, Thanavadee Prachason.

**Project administration:** Nida Limsuwan.

**Visualization:** Nitchawan Jongrakthanakij, Thanavadee Prachason.

**Writing – original draft:** Nitchawan Jongrakthanakij.

**Writing – review & editing:** Nitchawan Jongrakthanakij, Thanavadee Prachason, Komsan Kiatrungrit, Nida Limsuwan.

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
