## [Decision Letter · Decision Letter 0]

7 Dec 2025

Dear Dr. Prachason,

We look forward to receiving your revised manuscript.

Kind regards,

Gerard Hutchinson, MD

Academic Editor

PLOS One

Journal Requirements:

5. In the online submission form, you indicated that due to the inclusion of personal and sensitive information, the data from this study cannot be made publicly available. This restriction complies with the protocol approved by the Institutional Review Board (IRB) of Ramathibodi Hospital, Mahidol University. Data may be made available upon reasonable request and with approval from the IRB. Requests for data access should be directed to nitchawan.jon@mahidol.ac.th .

5. We notice that your supplementary [figures/tables] are included in the manuscript file. Please remove them and upload them with the file type 'Supporting Information'. Please ensure that each Supporting Information file has a legend listed in the manuscript after the references list.

Reviewers' comments:

Reviewer's Responses to Questions

**Comments to the Author**

1. Is the manuscript technically sound, and do the data support the conclusions?

Reviewer #1: Partly

Reviewer #2: Yes

2. Has the statistical analysis been performed appropriately and rigorously?

Reviewer #1: I Don't Know

Reviewer #2: Yes

3. Have the authors made all data underlying the findings in their manuscript fully available?

Reviewer #1: No

Reviewer #2: No

4. Is the manuscript presented in an intelligible fashion and written in standard English?

Reviewer #1: Yes

Reviewer #2: Yes

Reviewer #1: This is a very important area and I commend the authors for embarking on a study such as this. The integrative framework is a major strength of the study. However, there are a couple of important considerations I note below that may impact on the conclusions and implications stated by the authors.

1. The authors have not included a neurotypical control group- the reasons for this have not been elucidated.

2. The authors have also not considered the impact of comorbidities on family functioning and thereby on parental stress. The only consideration has been the child's ADHD symptoms - it would be important to consider the impact of comorbidities on family functioning and parental stress- if this is not able to be considered, at the very least, the reasons must be clarified and included in the limitations.

Reviewer #2: Thank you for giving me the opportunity to review the manuscript. I read it with great interest. The study investigated the role of child factors (ADHD severity and impairment) and family factors (parental ADHD severity and family functioning) in parental stress. Structural equation modelling revealed that child impairment, parental ADHD and family functioning are key factors in parental stress and should be addressed in multimodal ADHD treatment, rather than focusing solely on treating the child's ADHD symptoms. The abstract provides an informative overview of the study.

The introduction is clearly focused and includes rigorous argumentation to deduce the study's research question and the methods applied. The sample (recruitment and characteristics), instruments and statistics are all clearly described. The results are presented precisely and discussed adequately, including the study's limitations and clinical consequences. I only have a few minor suggestions for changes that could be made. The suggested changes are mandatory. It would also be possible to accept it as it is.

Abstract:

Methods:

Please consider adding basic sample characteristics such as age and gender.

p.1, l.16f: Please consider adding the notion that 'functional impairment' refers to the child (“Caregivers completed standardized measures of parental stress, child ADHD symptoms, child functional impairment, family functioning, and parental ADHD symptoms”).

Results:

Why are the statistical data for significant effects only presented for child- and family-related models, and not for the integrated model? Please consider presenting the data in the same way for all models, or for none of them.

Material and methods:

Why is inclusion criterion (3) restricted to the availability of data on parental stress? What about the other variables included in the equation modelling?

It would be helpful to add information on the diagnostic strategy applied in routine clinical care at the psychiatric outpatient department at Ramathibodi Hospital for assessing ADHD according to the DSM-5 and comorbid conditions.

Please provide information on the strategies employed to develop the Thai versions of the PSS-10 and ASRS-V1.1. Literature on the Thai versions of the other instruments is cited.

**Do you want your identity to be public for this peer review?** For information about this choice, including consent withdrawal, please see our Privacy Policy

Reviewer #1: No

Reviewer #2: No

---

## [Author Response · Author response to Decision Letter 1]

9 Jan 2026

Reviewer #1: This is a very important area and I commend the authors for embarking on a study such as this. The integrative framework is a major strength of the study. However, there are a couple of important considerations I note below that may impact on the conclusions and implications stated by the authors.

We sincerely thank you for your thoughtful and constructive feedback on our manuscript. We greatly appreciate your recognition of the importance of this area of research and your positive remarks regarding the integrative framework of our study. Your comments are invaluable in helping us improve both the clarity and interpretation of our findings. We have carefully considered your points and addressed them as follows:

1. The authors have not included a neurotypical control group- the reasons for this have not been elucidated.

R1.1: We thank the reviewer for raising this important point. This study utilized data derived from a clinical registry without including neurotypical families as a comparator because our primary aim was to examine the interrelationship between child-related and family-related factors associated with parental stress within families affected by ADHD, rather than to compare them with neurotypical families. However, we agree that the absence of a neurotypical control group limits direct comparison with normative levels of family functioning and parental stress. Therefore, we have now revised the Introduction to clarify the focus of our study on families affected by ADHD and have explicitly acknowledged the limitation of not having neurotypical comparators in the Discussion, as shown below.

Introduction

“…The present study aimed to address this gap by applying structural equation modeling (SEM) to examine the contributions of both child- and family-related factors to parental stress within families affected by ADHD. … Such insights would be a stepping stone to developing targeted, evidence-based interventions to mitigate parental stress and optimize treatment outcomes, not only for children with ADHD, but also for the entire family unit.”

Strengths and Limitations

“…Second, as this study aimed to explore the interrelationships between factors associated with parental stress within families affected by ADHD, we did not include neurotypical children as a control group, which limits the comparison of our findings with normative levels of family functioning and parental stress.”

2. The authors have also not considered the impact of comorbidities on family functioning and thereby on parental stress. The only consideration has been the child's ADHD symptoms - it would be important to consider the impact of comorbidities on family functioning and parental stress- if this is not able to be considered, at the very least, the reasons must be clarified and included in the limitations.

R1.2: Thank you for this thoughtful comment. We agree that comorbid conditions may affect family functioning and parental stress. However, the registry data used in this study were derived from a single assessment of routine psychiatric service, normally lasting 60-90 minutes; therefore, comorbidities that were not evident at the initial encounter may be overlooked or inadequately evaluated due to the time limit. As the prevalence of comorbid conditions commonly co-occurring with ADHD was low in our dataset, except for specific learning disorder, we conducted additional SEM sensitivity analyses adjusting for specific learning disorder, which also confirmed the main analysis. The results of these analyses were added to the revised manuscript as shown below, with detailed information in the supplement.

Result

Pathway from child ADHD symptoms to parental stress

“… Sensitivity analyses additionally controlling for family income or comorbid specific learning disorder confirmed the significant indirect effect of child ADHD symptoms on parental stress via functional impairment (S1 and S2 Fig)…”

Pathway from parental ADHD symptoms to parental stress

“.. Sensitivity analyses, additionally adjusting for family income and comorbid specific learning disorder, confirmed that both the direct and indirect effects remained significant (S4 and S5 Fig)…”

Integrated pathway from ADHD symptoms to parental stress

“...In sensitivity analyses, all models confirmed a significant indirect effect of parental ADHD symptoms on parental stress via family functioning, but not child functional impairment (S7-S9 Fig). However, the significant direct effect of parental ADHD was only confirmed in the model additionally adjusted for specific learning disorder (S8 Fig). The insignificant direct and indirect effects of child ADHD symptoms on parental stress through functional impairment remained consistent across all sensitivity analyses (S7-S9 Fig), but the indirect effect of child ADHD symptoms on parental stress through family functioning became significant in the model additionally adjusted for specific learning disorder (S8 Fig)…”

Furthermore, we have also added this point in the Limitation section as follows:

“Third, the registry data used in this study were derived from a single assessment of routine psychiatric service, normally lasting 60-90 minutes, with assurance of comprehensive clinical evaluation for ADHD diagnosis. Therefore, comorbidities that were not evident at the initial encounter may be overlooked or inadequately evaluated due to the time limit. Indeed, the prevalence of comorbid conditions commonly co-occurring with ADHD was low in our dataset, except for specific learning disorder. For this reason, we only additionally adjusted for specific learning disorder in our sensitivity analyses to confirm the robustness of the findings, rather than including it as a separate predictor in the models.”

Reviewer #2: Thank you for giving me the opportunity to review the manuscript. I read it with great interest. The study investigated the role of child factors (ADHD severity and impairment) and family factors (parental ADHD severity and family functioning) in parental stress. Structural equation modelling revealed that child impairment, parental ADHD and family functioning are key factors in parental stress and should be addressed in multimodal ADHD treatment, rather than focusing solely on treating the child's ADHD symptoms. The abstract provides an informative overview of the study.

The introduction is clearly focused and includes rigorous argumentation to deduce the study's research question and the methods applied. The sample (recruitment and characteristics), instruments and statistics are all clearly described. The results are presented precisely and discussed adequately, including the study's limitations and clinical consequences. I only have a few minor suggestions for changes that could be made. The suggested changes are mandatory. It would also be possible to accept it as it is.

We sincerely thank you for the time and effort dedicated to reviewing our manuscript and for providing such thoughtful and constructive feedback. We greatly appreciate your positive remarks regarding the focus, rigor, and clarity of our study, as well as the informative presentation of the abstract, methods, and results. Your insights have been extremely helpful in improving the clarity and completeness of our manuscript. We have carefully considered all your suggestions and addressed them as follows:

1. Abstract:

Methods:

Please consider adding basic sample characteristics such as age and gender.

R2.1: We thank the reviewer for this constructive comment. In response, the Abstract has been revised to include basic sample characteristics, including mean age and gender distribution, as shown:

“Methods: A cross-sectional study included 127 children and adolescents with ADHD (70.9% males; mean age 9.6 ± 3.3 years) and their caregivers, recruited from the ADHD Registry at Ramathibodi Hospital, Bangkok (2019–2023).”

2. p.1, l.16f: Please consider adding the notion that 'functional impairment' refers to the child (“Caregivers completed standardized measures of parental stress, child ADHD symptoms, child functional impairment, family functioning, and parental ADHD symptoms”).

R2.2: The wording in the Abstract has also been clarified to explicitly indicate that functional impairment refers to child functional impairment, as shown:

“Caregivers completed standardized measures of parental stress, child ADHD symptoms, child functional impairment, family functioning, and parental ADHD symptoms.”

3. Results:

Why are the statistical data for significant effects only presented for child- and family-related models, and not for the integrated model? Please consider presenting the data in the same way for all models, or for none of them.

R2.3: The Results section has been revised to ensure consistent reporting across all models, as follows:

“Results: Examining child- and family-related factors separately, child ADHD symptoms indirectly influenced parental stress via functional impairment, whereas parental ADHD symptoms significantly influenced parental stress both directly and indirectly via family functioning. In the integrated model examining both child- and family-related factors concurrently, the direct and indirect pathways from parental ADHD symptoms to parental stress via family functioning remained significant, but not the pathway from child ADHD symptoms to parental stress via functional impairment.”

4. Material and methods:

Why is inclusion criterion (3) restricted to the availability of data on parental stress? What about the other variables included in the equation modelling?

R2.4: Thank you for your thoughtful suggestion. This study utilized secondary data from the ADHD registry, a longitudinal cohort of children suspected of having ADHD, with the aim of investigating pathways to parental stress; therefore, the availability of this outcome variable was required for study inclusion. The sample with over 20% missing items of any questionnaire used in the models was excluded from the related analysis. To clarify these points, we have revised the Study design and participants section to make a clear distinction between the enrollment process of the ADHD registry and the inclusion and exclusion criteria for the current analyses. The justification for the inclusion criterion and the number of missing variables was now provided, as shown below.

“This cross-sectional study utilized baseline data from the ADHD Registry of the Department of Psychiatry at Ramathibodi Hospital, Bangkok, Thailand, a longitudinal cohort that consecutively enrolled children and adolescents suspected of having ADHD who first visited the psychiatric outpatient unit from October 2019 to October 2023. Children and adolescents who were unable to communicate in Thai were excluded. After informed written assent and consent were obtained from the participants and their parents, respectively, child and adolescent psychiatrists or psychiatry residents performed comprehensive clinical interviews with both the children and their caregivers to determine the diagnoses for the presenting problems as part of routine clinical care. A checklist of the DSM-5 criteria for ADHD [36] was recorded. The diagnosis of ADHD was defined as the presence of ≥6 symptoms in either the inattention or hyperactivity/impulsivity domains, beginning before age 12, present in at least two settings, and causing functional impairment. Children were excluded if symptoms occurred only during a psychotic disorder or were better explained by another psychiatric condition or substance use. Comorbidity identified at the initial visit was also recorded. Caregivers provided demographic and personal information (age, sex, living arrangements, marital status, education level, and family monthly income) and completed standardized questionnaires assessing parental stress, child ADHD symptoms, child functional impairment, family functioning, and parental ADHD symptoms. Multiple follow-up assessments were conducted over a one-year period after the initial visit. The Ethics Committee of Ramathibodi Hospital, Mahidol University, approved the data collection protocol of the registry (MURA2019/815).

For the present study, we retrieved anonymized baseline data from the ADHD registry comprising 164 children and adolescents who met the DSM-5 diagnostic criteria of ADHD. Only participants with available data on parental stress were included in this study because it was the primary outcome variable in the structural equation modeling. The sample with over 20% missing items of any questionnaire used in the models was excluded from the related analysis. The final sample for the main analyses comprises 127 participants. The characteristics of the excluded sample were not significantly different from the included sample (S1 Table). The Ethics Committee of Ramathibodi Hospital, Mahidol University, waived the requirement for informed consent to use the de-identified data for this study (MURA2025/340).”

5. It would be helpful to add information on the diagnostic strategy applied in routine clinical care at the psychiatric outpatient department at Ramathibodi Hospital for assessing ADHD according to the DSM-5 and comorbid conditions.

R2.5: Thank you for this helpful comment. We added clarification on the diagnostic strategy used in routine clinical care in the Method section as shown below.

Study design and participants

“…child and adolescent psychiatrists or psychiatry residents performed comprehensive clinical interviews with both the children and their caregivers to determine the diagnoses for the presenting problems as part of routine clinical care. A checklist of the DSM-5 criteria for ADHD [36] was recorded. The diagnosis of ADHD was defined as the presence of ≥6 symptoms in either the inattention or hyperactivity/impulsivity domains, beginning before age 12, present in at least two settings, and causing functional impairment. Children were excluded if symptoms occurred only during a psychotic disorder or were better explained by another psychiatric condition or substance use. Comorbidity identified at the initial visit was also recorded.”

6. Please provide information on the strategies employed to develop the Thai versions of the PSS-10 and ASRS-V1.1. Literature on the Thai versions of the other instruments is cited.

R2.6: We thank the reviewer for this helpful suggestion. We have now clearly referenced the original studies that developed and validated the Thai versions of the PSS-10 and ASRS-v1.1 with appropriate citations in the Measures section as follows:

“The Thai version PSS-10 has demonstrated good reliability and validity in Thai populations [38]…”

“The Thai version ASRS-v1.1 has been shown to be psychometrically reliable and valid for screening adult ADHD in Thai populations [51].”

---

## [Editor Report · Decision Letter 1]

13 Jan 2026

From ADHD symptoms to parental stress: The roles of functional impairment, family functioning, and parental ADHD

PONE-D-25-61073R1

Dear Dr. Prachason

We’re pleased to inform you that your manuscript has been judged scientifically suitable for publication and will be formally accepted for publication once it meets all outstanding technical requirements.

Kind regards,

Gerard Hutchinson, MD

Academic Editor

PLOS One
---

## [Editor Report · Acceptance letter]

PONE-D-25-61073R1

PLOS One

Dear Dr. Prachason,

I'm pleased to inform you that your manuscript has been deemed suitable for publication in PLOS One. Congratulations! Your manuscript is now being handed over to our production team.

Kind regards,

on behalf of

Dr. Gerard Hutchinson

Academic Editor

PLOS One